# Differences in Perception of Healthcare Management between Patients and Professionals

**DOI:** 10.3390/ijerph20053842

**Published:** 2023-02-21

**Authors:** Diego Moya, Mercedes Guilabert, Rafael Manzanera, Gloria Gálvez, Marta Torres, Adriana López-Pineda, María Lourdes Jiménez, José Joaquín Mira

**Affiliations:** 1Healthcare and Prevention Services Area, MC Mutual, 08037 Barcelona, Spain; 2Health Psychology Department, Miguel Hernández University, 03202 Elche, Spain; 3Independent Researcher, 08172 Barcelona, Spain; 4Clinical Medicine Department, Miguel Hernandez University, 03550 Sant Joan d’Alacant, Spain; 5Atenea Research Group, Foundation for the Promotion of Health and Biomedical Research, 03550 Sant Joan d’Alacant, Spain; 6Department of Emergency Medicine, College of Medicine, University of the Philippines, Manila 1500, Philippines; 7Alicante-Sant Joan d’Alacant Health Department, 03013 Alicante, Spain

**Keywords:** quality of health care, patient satisfaction, health care professionals, patient outcome assessment, patient-centered care, mutual insurance company, health care economics and organizations

## Abstract

Patient perception and the organizational and safety culture of health professionals are an indirect indicator of the quality of care. Both patient and health professional perceptions were evaluated, and their degree of coincidence was measured in the context of a mutual insurance company (MC Mutual). This study was based on the secondary analysis of routine data available in databases of patients’ perceptions and professionals’ evaluations of the quality of care provided by MC Mutual during the period 2017–2019, prior to the COVID-19 pandemic. Eight dimensions were considered: the results of care, coordination of professionals, trust-based care, clinical and administrative information, facilities and technical means, confidence in diagnosis, and confidence in treatment. The patients and professionals agreed on the dimension of confidence in treatment (good), and the dimensions of coordination and confidence in diagnosis (poor). They diverged on confidence in treatment, which was rated worse by patients than by professionals, and on results, information and infrastructure, which were rated worse by professionals only. This implies that care managers have to reinforce the training and supervision activities of the positive coincident aspects (therapy) for their maintenance, as well as the negative coincident ones (coordination and diagnostic) for the improvement of both perceptions. Reviewing patient and professional surveys is very useful for the supervision of health quality in the context of an occupational mutual insurance company.

## 1. Introduction

Since the 1960s [1], patient satisfaction has been considered an indirect indicator of quality of care, and its measurement is present in all quality models [2]. Patient satisfaction can result in an increase in therapeutic compliance, less use of health resources, and better prognosis [3,4]. Patient satisfaction can be determined by socio-demographic factors, the healthcare system or center’s characteristics, and patient expectations, among other factors [5].

Meanwhile, work environments, organizational and safety cultures, and the well-being of health professionals have been associated with the quality of care [6]. Although professional satisfaction has been considered a precursor of quality of care, it was not until the quadruple-aim model for healthcare (patient, community, cost, and professional) was provided by the Institute for Healthcare Improvement (IHI) in the first decade of the 21st century that it acquired a more significant position among the favorable strategies for ensuring quality of care [7]. Determinants, both at an individual level, such as age or attitude, and at the organizational levels, such as structure or dynamics, contribute to healthcare professionals’ well-being [8].

The attitudes and expectations of patients and health professionals, and the communication between them, can be determinants of the final product in the health service companies [9]. Therefore, quality policies must consider the experience of patients, the climate and culture of an organization, and the professional well-being of professionals. For the measurement of this set of experiences, organizational and socio-labor factors can offer clues about the degree of implementation of the quality and safety policies of health institutions and about the level of achievement of these initiatives [7]. Some studies have demonstrated that the agreement on patient care management perceptions between health professionals and patients only sometimes coincides, and that the satisfaction of the professionals does not necessarily affect the satisfaction of patients [10,11]. Patients and professionals could evaluate quality of care differently according to their knowledge and responsibilities [12]. Thus, in aspects related to access to consultation, patients think, contrary to health professionals, that the appointment system limits access, and that the waiting time from when they need to contact their doctor until they have the consultation is too long [13]. Patients and professionals have differing views on their contact with the family doctor by mobile phone, although professionals also highlight the need for protocols for telephone consultations [14].

Regarding home consultations, patients believe this action provides greater comfort and safety; however, professionals show a much more restrictive view, with few believing that it is necessary [15,16]. Additionally, there are differences between the perceptions of patients and healthcare professionals of teamwork in healthcare. Pullon et al. [17] observed that, while professionals perceived themselves as working in healthcare teams with defined roles, patients seemed unaware of each other or their roles. In addition, the patients believed their doctor had leadership qualities, while the doctor thought this leadership was shared. Meanwhile, the patients did not seem to recognize nurses’ abilities as coming from real professionals, and therefore did not believe nurses could make decisions [17].

Surveys among patients and professionals could be an element to deepen this analysis and to concretize the values we use for our organization. Data on the PREMS (service provided) and PROMS (individual experience) measures provide critical information for developing patient-centered care. They reduce the observed gap between the vision of the physician and of the patient and design treatment plans to meet the patient’s preferences and needs [18]. Some surveys allow the approximation of PROMS and PREMS through questions addressed to patients and professionals [19].

The mutual collaborators of social security (MCSS) in Spain predate the current public health systems and cover the risks of occupational disease, disability, invalidity, and death concerning work activity [20]. MCSS in Spain conducted comprehensive management of professional contingencies (i.e., work accidents and occupational diseases), which included carrying out preventive activities for labor accidents and offering healthcare to the injured in a personalized form of the highest quality of care, including rehabilitation, readaptation, and reintegration. They also manage the procedures and payment of economic benefits to injured and sick workers. The disability coverage ranges from non-invalidating permanent injuries to absolute permanent disability with all grades in between. A worker’s death, loss of spouse and orphans are also covered.

In order to consolidate the quality of care, MCSS follow a strategy to obtain the certifications of the ISO 9001, the quality model of the European Foundation for Quality Management (EFQM) [21], the quality accreditation and healthcare (QH) model. In 2010, the UNE 179003 standard was introduced to improve the distinct aspects of patient care that affect safety, which paved the way for insurance companies to begin safety programs [22] to achieve excellence and provide services in the healthcare field for occupational accidents and diseases.

Health quality programs must include the evaluation of patient satisfaction as the central objective of their care activities, and Health quality programs must include the evaluation of patient satisfaction as the central objective of their care activities. They must also include the opinion of professionals as the main guarantee of the quality of health initiatives [23]. The MCSS’s opinions and assessments of both are measured systematically through patient surveys and analyses of the safety culture of professionals [24,25]. In labor insurance, there are different circumstances to those that occur in conventional hospitals and health centers, since in insurance companies and doctors also make decisions about workers’ ability to remain at work with significant economic repercussions. Situations of illness and injury affect workers’ salary and, therefore, their expectation of economic income. This fact affects patients’ perceptions of the care received and the results of said care.

Knowing the coincidences and differences between the evaluations completed by professionals and patients of an MCSS company of perceived care could help the insurance company to identify strengths and weaknesses in the levels of quality of care and patient safety and to develop new strategies for improvement. Therefore, the main objective of this study is to identify, using common instruments (patient and professional culture surveys) used by occupational health insurance companies involving external and internal clients of the organization, common elements to find points of convergence and divergence in three years (2017, 2018, and 2019) prior to the COVID-19 pandemic.

## 2. Materials and Methods

This study is based on a secondary analysis of the routine data available in databases of patient satisfaction and professional evaluations of the quality of care provided by MC Mutual during 2017–2019. MC Mutual conducted this study last 2021, which was approved by the Project Evaluation Board of Miguel Hernández University with code DPS approved.JJM.03.20.

MC Mutual is a private non-profit organization financed by the social security system that covers the needs of more than 177,000 companies and 1,500,000 workers. The study considered professional consultations (health care) and 2983 surgical interventions during the 2017–2019 period. At this point, there were 857 health professionals, two hospitals (Clínica Copérnico and Clínica Londres) with 75 beds in total, and 100 outpatient centers, 88 of which dealt with accidents at work and occupational diseases, and the assessment of common diseases that receive social security benefits was considered as part of the evaluation.

In 2013, MC Mutual launched a Health Quality Plan, which included patient safety as one of its priorities. The plan has been implemented up to the present and considers two main protagonists: patients, the center of care and evaluation initiatives, and professionals, the main guarantee of the quality of health initiatives. The opinions and assessments of patients and professionals have been measured systematically through patient surveys and the quality program for professionals, focusing on the deployment of the successive quality plans [26].

Figure 1 shows the study phases carried out to establish the process of comparing opinions between patients and professionals of MC Mutual.

The patients and professionals correspond to the resources and activity in the 88 care centers of MC Mutual. No survey has been excluded for any reason.

### 2.1. Phase 1: Study Selection Period and Data Collection

#### 2.1.1. Patients’ Point of View

Since 2016, patients’ opinions have been assessed annually through satisfaction surveys incorporating patient voice in evaluating care experience. The first surveys conducted in this study measured the satisfaction of patients treated with professional consultations. For inpatients, the evaluation was administered physically to the patient at the time of medical discharge. The aspects evaluated were the information received, the treatment, waiting time, and facilities. For outpatients, the surveys were administered to patients by telephone shortly after being seen. These surveys, included the evaluation of the following aspects: information received, the treatment, whether the information could be flexible and adaptable, conformity with diagnoses and treatments, the comfort provided by the facilities, the accessibility and proximity of the centers and the agility allowed between them, and the waiting times for services. In the case of the outpatient centers, the assessment of work-related care was conducted. At the clinics, the services that the patients evaluated were hospitalization, surgery, rehabilitation, external consultations, and psychology, using the same dimensions as those in outpatient centers. The survey response scale included scores from one to ten, where one meant very dissatisfied and ten meant very satisfied [26]. The years 2017–2019 were been selected, within which the surveys were carried out in the healthcare centers.

#### 2.1.2. Professionals’ Point of View

In 2014, professional perception studies began with two complementary approaches: Quality Antenna (specific topics) and Quality Culture (a general approach).

The Quality Antenna approach was used to explore the opinions of professionals on specific aspects of the initiatives that the Health Quality Plan proposed to address, in relation to the patients, continuity of care, diagnostic and therapeutic adequacy, technical competence, job satisfaction, accessibility, clinical safety, and equity. Surveys were conducted using adapted questions on five successive Health Quality Plan consultations (to assess the level of the development of the quality initiatives defined in each of the plans). Table A1 highlights the elements that the survey identified throughout the period 2017–2019 and other questions asked in some periods. The Quality Antenna survey collected the opinions of MC Mutual professionals from both large centers (those with more than ten health professionals and more than 5000 cases requiring medical attention per year) and those considered small centers (those with less than four professionals for health services and less than 1500 cases requiring medical attention per year The opinions were collected during the study period from groups health professionals. The number of individuals included in these groups ranged between 150 and 170. Likewise, with the same survey, the opinions on the 87 Quality Referents (CAS, “Calidad ASistencia” in Spanish) were collected, and these refer to professionals designated in each care center, agreed by managers and health professionals of the center, who ensure compliance and knowledge of the Plan of Sanitary Quality.

### 2.2. Phase 2: Generation of Categories and Analysis of Concordance

#### 2.2.1. Generation of Categories by Members of the Research Group

The surveys distributed to professionals and those distributed to patients were different, as shown in Table A2, but their items are based on the same dimensions of quality of care. We considered these dimension categories, which allowed us to carry out this study. To compare the opinion of patients and professionals, the research group was constructed of nine categories and their corresponding operational definitions (Table 1). In generating the first grid, a group of experts validated the degrees of agreement according to the classification proposed by the research group. To carry out this comparative analysis, the number of items in the opinion studies of patients and that in the opinion studies of professionals s was considered different. However, they were classified according to the definitions of the categories.

#### 2.2.2. Concordance Analysis

To compare the results of the items addressed to the patients versus the results of the items addressed to the healthcare professionals, the assignments made for the items of both instruments were validated by external experts. A group of three external experts (two clinical experts in quality and an academic expert in quality and patient safety) was asked to assess the degree of agreement between items within the same category of the patient opinion survey and of the Quality Antenna survey of professionals. The experts scored the association between these items using a scale of 1 to 5 points, from lowest to highest agreement.

With the scores obtained by external experts, the Kendall coefficient of concordance (W) [27] was calculated. The Kendall coefficient result was interpreted using the classification established by Landis and Koch [28] to describe the consistency associated with the statistics with a range of agreements ranging from poor to perfect (the fair agreement values of 0.20 to 0.40 and the moderate agreement values of 0.40 to 0.60 were considered).

### 2.3. Phase 3. Assignment of Nominal Order of Professionals, Referents, and Patients

Finally, to establish the comparison, the following steps were taken. First, it was necessary to standardize the scores of the two surveys on a scale of 1 to 5 points; a regression line was determined that allowed the scores to be transformed and made comparable (Y: 5/9 + 4/9 × X). Data from both surveys used the same scale and integrated the scores into a summary table of the three groups. We used diachronic analyses to formulate the evolutionary trend of each group of patients, professionals, and referents. Later, an order was assigned to the average of the scores of professionals, referents, and patients, allowing the comparative analysis. Spearman’s ordinal regression coefficient was used to compare the ordinal values of the professionals and referents first, and then it was to compare those of the professionals (including referents) and patients. Finally, the ordinal values were presented (the first value is the best and higher numbers are worse) and a graph of quadrant coincidences and divergences between the professionals (including the referents) and the patients was elaborated.

## 3. Results

The surveys sent to the professionals had a response rate of 40% (*n* = 171), 51% (*n* = 154), and 37% (*n* = 150) in 2017, 2018, and 2019, respectively. In the case surveys sent to healthcare quality referents, the response rate was 70% (*n* = 89), 87% (*n* = 87), and 67% (*n* = 87) in 2017, 2018, and 2019, respectively. Out of the 107,468, 110,683 and 115,069 annual patients who were in outpatient care, respective to the three years, 2950 (2.7%), 2674 (2.4%), and 2646 (2.3%) responded to the patient surveys.

### 3.1. Results of the Inter-Rater Agreement Analysis

In 2017, the degree of expert agreement was W = 0.361 (median degree of agreement); in 2018, the result was W = 0.622 (good degree of agreement). In 2019, the agreement between experts was W = 0.529 (moderate degree of agreement). Although there was variability between the results, all the values obtained indicated that agreement was met. Specifically, there was agreement between the experts regarding the distribution of items concerning the categories generated by the research team. Specific categories, such as treatment referrals, lack of continuity of care, and information overload, were eliminated from the surveys (to referents and professionals) since there was low agreement between the experts.

### 3.2. Results in the Study Period in Referents, Professionals, and Patients

Table 2 shows the average score by category of analysis for each of the periods and comparison groups. The scores of the categories were similar in professionals and referents and were grouped at around 3.5 points, and the patients valued all the categories better. According to the referents, the temporal evolution was stable, except in facilities where the scores followed an upward trend.

Table A2 shows the results in each analysis period for each category. The items that made up each category are also shown with the scores for the patients, professionals, and referents.

### 3.3. Coincidences and Divergences between Referents, Professionals, and Patients

The values observed in the patients, referents, and professionals obtained across the three years considered were averaged (Table 3). When ordering the values, there were no differences between referents and professionals, and a high correlation between the values of each was shown (Spearman’s Rho 0.84, *p* < 0.01). For this reason, we grouped, as an average, the values for professionals and for referents in Table 4, calculating their ordinals again. We reviewed the differences from a statistical point of view (Spearman’s Rho 0.09 *p* > 0.05 not significant). The values furthest from their ordinals were for care results, care with confidence, facilities, treatment, and medical information.

Figure 2 of the coincidence and divergence quadrants shows the coincidences between professionals (including professionals and referents) and patients who negatively assessed coordination and diagnosis and who positively assessed treatment. The upper right-hand corner (in green) represents the coincidence in the categorization of the most highly rated health service by both groups. The bottom left (in red) shows, on the other hand, the coincidences in the worst ratings. The results about trust and treatment showed that professionals scored this variable higher than patients. In contrast, the patients valued the facilities, the results, and the information that the professionals gave them more than the professionals did, according to the opinions these professionals expressed.

The results showed that the methods commonly used to measure the perceptions of patients and professionals were practical but expressed different assessments of care. In the most subjective aspects (trust and treatment), the patients had worse perceptions concerning the health professional. The more technical aspects (facilities, results, and medical information) were better valued by patients than professionals. There was an agreement between professionals and patients in treatment (good), and coordination and diagnosis (bad).

## 4. Discussion

The orientation towards values in health initiatives is based on the clinical improvement vocation of professionals within the framework of institutional support but requires a specific analysis and evaluation methodology. Quality care improvement involves the adaptation of guidelines and protocols, and the organization of the needs of homogeneous groups of patients is an essential element. All of this is within clinical management units’ clinical leadership frameworks. A change from considering the patient as object to considering the patient as subject and a consideration of how the health intervention affects the patient’s perceptions are essential to confirm the quality of the process and the result of the health intervention [29].

In our case, the perceptions of satisfaction in the different dimensions of health quality were similar between professionals and referents, which suggests that training and loyalty strategies should be reinforced among health quality referents in healthcare centers. This reference figure should act as a local catalyst, helping to improve the perceptions of their healthcare colleagues and contributing to improving the opinions of all professionals.

Some studies that combine patient and professional preferences in the design of ambulatory care [30], communication in the consultation [31], and the care of elderly patients [32], which began a long journey in the field, are classic examples of combinations of patient and professional preferences for improving patient-centered care. Professionals may become more aware of Health Quality Plan strategies when they observe (as this study shows) that patients need more confidence in them and better treatment, even though they do not perceive them as deficient.

On the contrary, the treatment results show that the information these professionals offer and the facilities available are valued more by the patients than the professionals. Professionals are aware of the workloads they suffer at certain times and the technological deficits that may exist.

Some studies have investigated which elements are critical to the perception of patients. Waiting and resolution capacity are especially critical. In emergencies, adequate face-to-face contact, patient location, attention to emotional disturbances, and permanent telephone contact are recognized as essential [33].

Coordination is difficult to define in terms of perception, but it suggests that it may occur in the form of work peaks (which are impossible to foresee), a feeling of lack of time or confidence. The Quality Plan must be more proactive in these dimensions since patients and professionals valued them poorly.

In Korean care centers, different perceptions were detected between nurses and patients. The nurses rated the health care worse than the patients did. Expectations played a crucial role in these differences and must be considered [34]. The same was found to be true in the case of perceptions of patients and nurses of treating postoperative pain, with substantial disagreements being revealed over whose analysis improves the provision of these services in Swedish clinics [35].

Our study suggests that professionals and patients agree that treatment is adequate and that its results are appropriate. The Health Quality Plan, built in a participatory manner, insists on some initiatives that are based on the reality of our patients and our clinical capacities. PREMS and PROMS are good instruments for dialogue and clinical management between patients and professionals. The surveys allowed us to have a good approximation of this discussion, which revealed coincidences between both protagonists in their appreciation of the medical attention given.

Reviewing patients’ and professionals’ norms regarding the provision of information to patients in French acute care hospitals, no differences were found in any of the opinions explored between professionals and patients [36]. In contrast, our work found some aspects that coincided (treatment being valued positively and coordination and diagnosis being valued negatively), although in other aspects, the disagreement was significant (results and information were more highly valued by patients; respect and trust were more highly valued by professionals).

Studies conducted in primary care have shown a strong correlation between the perceptions and expectations of patients and professionals regarding the importance of accessibility and continuity of consultations [37]. Unlike in other specialized fields, such as oncology, neurology, or cardiology, there is a well-founded reluctance towards indiscriminate genetic care within primary care [38]. However, other studies highlight the concern about access and the cost of this type of service in the USA [39].

The collaboration between patients, professionals, and administrators is established based on information from the hospital centers of a district in Finland. According to all stakeholders, the improvement in information flows and the greater involvement of patients in centers improves patient-centered care [40].

Some pathologies, such as fibromyalgia, are particularly complex to diagnose and sensitive to address within the patient–professional relationship. They and have been the subject of studies on coincident and divergent aspects. Moreover, the perceptions of the diagnosis and treatment of this disease among patients and professionals has made it possible to improve professional support and the adaptation of health resources to the patients [41]. In our case, something similar could happen, particularly in patients with the most severe work accidents, which are those that will alter their functional capacity. According to other studies, something similar happens to affected patients when sensitivity to the pathology is considered, as it does to relatives of those affected by hereditary colorectal cancer when they share their concerns with primary care physicians [42].

Even though the institutional collaboration strategies promoted by the Institute for Clinical System Improvement proved less valuable than expected, they must still be insisted. The proposed move will support the development of the organizational climate for quality improvement and quality of service, and with these, the better perception of the care actions taken for the patient (waiting, communication, and recommendation of the service) [43]. Subsequent studies closer to our cultural reality clearly show that most patients prefer the decisions to be made by their doctor, especially in the cases of patients with more severe illnesses and older patients. The information provided by general practitioners from their own point of view does not seem to contribute enough to patient involvement in the clinical process [44].

The quadruple aim model [45] supports the idea that the well-being of professionals is essential in achieving excellent quality care results. This aspect is a prerequisite for achieving optimal quality care, increasing communication between patients and professionals, and sharing decisions, and will be an essential guide for our future concrete and strategic actions.

### Limitations

The data came from a labor mutual insurance company and must be considered specific to this environment, so the comparisons must be used with caution. However, myriad studies in different environments are mentioned in the work.

Most mutuals collaborating with the Spanish Social Security system systematically carry out opinion studies of patients and professionals, which invites us to confirm the conclusions in other entities like ours before concluding their applicability and generalizability in other entities.

The data came from systematic surveys; therefore, they are qualitative and comparison data. A transformation of the data was conducted to make the categories of analysis comparable. An attempt was made to minimize subjectivity by having a panel of experts show the degree of agreement among them on the proposed categorization.

The data refer to the pre-pandemic period, so the situation should be reviewed once the pandemic is over.

Despite the limitations of the study, it is worth highlighting the strengths which have been found: One such strength is that it identified a minimum set of categories that were necessary to compare the viewpoints of two types of stakeholders in a healthcare organization. On the other hand, this was a pioneering study as far as this type of organization was concerned, providing aspects from two main actors in the care process, which gives visibility to the elements which should be incorporated into future strategic plans in terms of quality and patient safety.

## 5. Conclusions

The review of patients’ and professionals’ survey is a beneficial element for supervising the health quality provided by our proposed actions making decisions to improve the care according to them priorities, perceptions, and needs.

The patients who are treated at the mutual insurance company have a special characteristic, different from the usual care in health centers. In this case, the professional’s decision involves not only clinical decisions (as usual) but also a decision on their employment situation (active or off work) and, therefore, affects their salary. By comparing the perspectives of both in this study, we provide qualitatively different information to understand key factors of the patient-healthcare provider encounter.

Nevertheless, a good instrument for ensuring transparency will be valued positively by our members: the companies that constitute the entity of concern in our study; the public entity that protects us, social security; the workers’ representatives; the unions that constitute the critical tripod that sustains the collaborating mutuals. The future effort put in by our managers and professionals to pursue the quadruple goal will favor the institutional tripod that supports us.

The care managers must reinforce the training and supervision activities of the positive coincident aspects (therapy) for their maintenance, as well as the negative coincident ones (coordination and diagnostic) for the improvement of both perceptions.

The study encourages us to review the PREMS and PROMS measures aimed at the most critical points of disagreement related to trust and treatment, where the patients had worse perceptions regarding the health professional, or those aimed at aspects such as medical information, which was better valued by the patients than by the professionals. It is also recommended that coordination and diagnosis are delved into, since these were considered aspects that could be improved by both groups. All these considerations constitute essential elements in future healthcare quality strategies for our organization.

## Figures and Tables

**Figure 1 ijerph-20-03842-f001:**
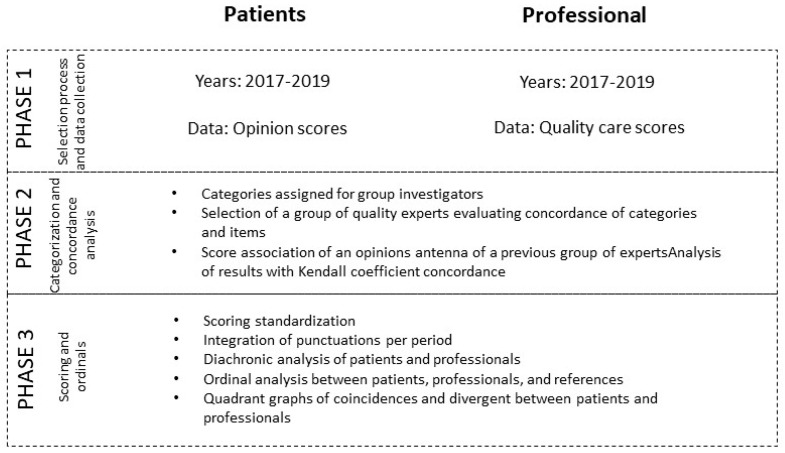
A three-phase methodological approach to studying patient opinions and professional culture.

**Figure 2 ijerph-20-03842-f002:**
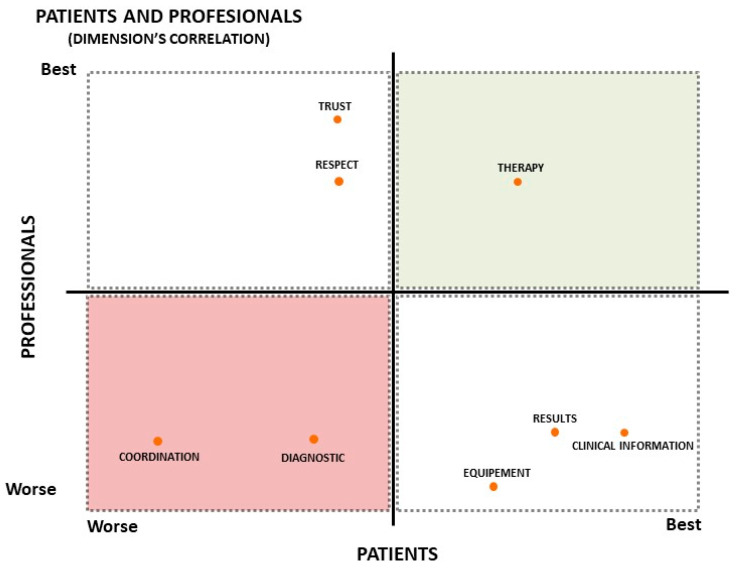
Quadrants of coincidences and divergences between professionals and patients. The red dots express the perceptions of patients and professionals. The red quadrant contains the categories worst rated by both. The green quadrant contains the categories most highly rated by both.

**Table 1 ijerph-20-03842-t001:** Evaluation of health quality categories in the questionnaires used for patients and professionals.

Categories	Definitions
Global evaluation of care	Global assessment of care
Clinical outcomes	Best outcome
Coordination between professionals	Everything is functioning well
Trust based healthcare	I trust in the entire healthcare process
Clinical and administrative process	I have understood all the information that I needed
Installations and technical resources	Space and technical resources are adequate
Confidence in the diagnostics	The test has clear indications in my situation
Treatment	I have been treated well
Confidence in the treatment	I have received adequate treatment in my situation

**Table 2 ijerph-20-03842-t002:** Scoring of the categories in the study period of referents, professionals, and patients.

	2017			2018			2019		
Categories	Referents (*n* = 62)	Professionals (*n* = 69)	Patients (*n* = 2950)	Referents (*n* = 76)	Professionals (*n* = 78)	Patients (*n* = 2674)	Referents (*n* = 58)	Professionals (*n* = 55)	Patients (*n* = 2646)
Treatment results	3.6	3.6	4.5	4.0	3.7	4.4	3.9	3.8	4
Coordination between professionals	3.4	3.2	4.5	3.5	3.5	4.3	3.6	3.5	4.2
Trust based healthcare	3.6	3.7	4.6	3.6	3.6	4.5	3.5	3.5	4.8
Clinical and administrative process	4.3	4.3	4.5	4.3	4.2	4.4	-	-	4.2
Installations and technical resources	2.8	3.7	4.5	3.3	3.2	4.5	3.5	3.4	4.4
Confidence in the diagnostics	3.8	3.7	4.6	3.5	3.5	4.4	3.5	3.5	4
Respect	3.8	3.5	4.6	3.6	3.5	4.5	-	-	4.4
Confidence in the treatment	3.7	4.0	4.6	3.8	3.6	4.5	3.9	3.7	4.4

*n* = referents, professionals, and patients surveyed. The “-” markings indicate that data were not available because they had been removed from the survey for its stability.

**Table 3 ijerph-20-03842-t003:** Average scores of referents, professionals, and patients and ordinal data.

Categories	Referents 2017–2019 (*n* = 196)	Professionals 2017–2019 (*n* = 202)	Patients 2017–2019 (*n* = 8270)	Ordinal for Referents	Ordinal for Professionals	Ordinal for Patients
Treatment results	3.8	3.7	4.3	2.5	3	6.5
Coordination between professionals	3.5	3.4	4.3	7	7.5	6.5
Trust based healthcare	3.6	3.6	4.6	5.5	4.5	1
Clinical and administrative process	4.3	4.3	4.3	1	1	6.5
Installations and technical resources	3.2	3.4	4.5	8	7.5	3
Confidence in the diagnostics	3.6	3.6	4.3	5.5	4.5	6.5
Respect	3.7	3.5	4.5	4	6	3
Confidence in the treatment	3.8	3.8	4.5	2.5	2	3

For the ordinals, the best quality perceived is represented by lower numbers (green), and those ordered as higher numbers represented the worse perceived quality (orange or red). Green: order between first and third. Orange: order between fourth and fifth. Red: order between sixth and higher.

**Table 4 ijerph-20-03842-t004:** Average scores between professionals and referents and patients.

Categories	Referents and Professionals 2017–2019 (*n* = 398)	Patients 2017–2019 (*n* = 8270)	Ordinal for Referents and Professionals	Ordinal for Patients
Treatment results	3.8	4.3	2	6.5
Coordination between professionals	3.4	4.3	7	6.5
Trust based healthcare	3.6	4.6	5	1
Clinical and administrative process	4.3	4.3	1	6.5
Installations and technical resources	3.3	4.5	8	3
Confidence in the diagnostics	3.6	4.3	5	6.5
Respect	3.6	4.5	5	3
Confidence in the treatment	3.7	4.5	3	3

For the ordinals, the best quality perceived is represented by lower numbers (green), and those ordered as higher numbers represented the worse perceived quality (orange or red). Green: order between first and third. Orange: order between fourth and fifth. Red: order between sixth and higher. The compilation of the data from the different rounds and surveys related to patients and professionals is shown in Table 4. The average ratings indicated a favorable opinion among professionals (ranging from 3.3 to 4.3) and patients (4.3 and 4.6) (Table 4).

## Data Availability

The full dataset and statistical code are available from the corresponding author.

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
