# Peer review of "Differences in Perception of Healthcare Management between Patients and Professionals"

_ijerph, 2023, doi:10.3390/ijerph20053842_

Round 1

Reviewer 1 Report

This is an interesting study based on patients' perceptions and professionals' evaluations of the quality of care. Eight dimensions were considered in the evaluation.  The weakness of the study that it does not use standardized validated scales. Nevertheless, the results are important and worth publication. I recommend this paper to be accepted after the language editing by a native speaker. There are some grammatical errors in the abstract and in the text of the manuscript.

Author Response

Reviewer 1

This is an interesting study based on patients' perceptions and professionals' evaluations of the quality of care. Eight dimensions were considered in the evaluation.  The weakness of the study that it does not use standardized validated scales. Nevertheless, the results are important and worth publication. I recommend this paper to be accepted after the language editing by a native speaker. There are some grammatical errors in the abstract and in the text of the manuscript.

Comments:

1.1. This is an interesting study based on patients' perceptions and professionals' evaluations of the quality of care. Eight dimensions were considered in the evaluation. 

Response: We are very grateful for the positive feedback of the referee.

1.2. The weakness of the study that it does not use standardized validated scales. Nevertheless, the results are important and worth publication.

Response: We appreciate the suggestion. We are very grateful for the positive feedback in relation to the publication.

1.3. I recommend this paper to be accepted after the language editing by a native speaker. There are some grammatical errors in the abstract and in the text of the manuscript.

Response: We appreciate the suggestion. Accounting for the given suggestion, the manuscript has been revised by a native speaker and the grammatical errors have been corrected.

Reviewer 2 Report

This is an interesting study that assesses the differences and coincidences between the perception of care by professionals and patients in a mutual health insurance company.
In my opinion, the work highlights a relevant health issue of interest to the scientific community, especially for professionals working in the field of quality.
Although the article is well thought out and developed, there are aspects that should be clarified and modified before publication.
I have classified them as minor changes, as they are not significant methodological errors that invalidate the study, but I understand that they are relevant and should be addressed.
I have incorporated my contributions, in commentary format, in the attached document.
Best regards

Author Response

Reviewer 2

This is an interesting study that assesses the differences and coincidences between the perception of care by professionals and patients in a mutual health insurance company. In my opinion, the work highlights a relevant health issue of interest to the scientific community, especially for professionals working in the field of quality. Although the article is well thought out and developed, there are aspects that should be clarified and modified before publication. I have classified them as minor changes, as they are not significant methodological errors that invalidate the study, but I understand that they are relevant and should be addressed. I have incorporated my contributions, in commentary format, in the attached document. Best regards

2.1. This is an interesting study that assesses the differences and coincidences between the perception of care by professionals and patients in a mutual health insurance company.

Response: We are very grateful for the positive feedback of the referee.

2.2. In my opinion, the work highlights a relevant health issue of interest to the scientific community, especially for professionals working in the field of quality. Although the article is well thought out and developed, there are aspects that should be clarified and modified before publication.

Response: We are very grateful for the positive feedback. We appreciate the suggestion. Accounting for the given suggestion, we have addressed all the aspects mentioned below.

2.3. I have classified them as minor changes, as they are not significant methodological errors that invalidate the study, but I understand that they are relevant and should be addressed. I have incorporated my contributions, in commentary format, in the attached document. Best regards

2.3.1. Introduction

2.3.1.1. You should explain what the IHI quadruple aim is and what the initials IHI stands for (line 46)

(…) it was not until the definition of the Institute for Healthcare Improvement (IHI) quadruple aim for healthcare model (patient, community, cost and professional) in the first decade of the 21st century (…)

2.3.1.2. Citation number 10 should go at the end of the sentence [10, 11] (line 57)

Response: We appreciate the suggestion. Accounting for the given suggestion, the position in the text of reference number [10] is modified.

(…) Some studies demonstrated that the agreement on the perceptions of agreement of patient care management perceptions between health professionals and patients only sometimes coincides, and that the satisfaction of the former does not necessarily affect the satisfaction of patients [10, 11].

2.3.1.3. You should add citations to support this view. (lines 65 and 66)

Response: We appreciate the suggestion. Accounting for the given suggestion, a reference (Ball et al.) is added to support the reflection.

Patients and professionals have differing views on their contact with the family doctor by mobile phone, although professionals also highlight the need for protocols for telephone consultations [14].

[14]: Ball SL, Newbould J, Corbett J, et al. Qualitative study of patient views on a ‘telephone-first’ approach in general practice in England: speaking to the GP by telephone before making face-to-face appointments. BMJ Open 2018;8:e026197. doi:10.1136/ bmjopen-2018-026197

2.3.1.4. You should add citations to support this view. (lines 68 and 69)

Response: We appreciate the suggestion. Accounting for the given suggestion, two references (Theile et al. and Smith-Carrier et al.) are added to support the commentaries.

Regarding home consultations, patients believe this action provides greater comfort and safety; however, professionals show a much more restrictive view, while with a few believing that it is necessary [15,16].

[15]: Theile et al. BMC Family Practice 2011, 12:24 http://www.biomedcentral.com/1471-2296/12/24 (22 April 2011)

[16]: T. Smith-Carrier et al. It ‘makes you feel more like a person than a patient’: patients’ experiences receiving home-based primary care (HBPC) in Ontario, Canada. Health and Social Care in the Community (2016).

2.3.1.5. The validity and, if possible, the psychometric properties of the questionnaires used should be described in the methodology section. (line 107)

Response: We appreciate the suggestion. Accounting for the given suggestion, changes are incorporated into the text to improve the meaning of the sentence. Also, during this commentary revision and addressing that the description must be in the methodology section, an error has been detected in the References section (reference 22 in text, reference 21 in References section). This reference is now number 26 and located in the Material and Methods section. This reference discusses in detail the validation conditions of the instruments.

(…)Therefore, the main objective of this study was to identify, using the traditional common instruments (patient and professional culture surveys) by occupational health insurance companies involving external and internal clients of the organization, common elements to find points of convergence and divergence in three years of study (2017-2018-2019) prior to the COVID-19 pandemic.

In 2013, MC Mutual launched a Health Quality Plan, which included patient safety as one of its priorities. The plan has been implemented up to the present, considering two main protagonists: patients, as the center of care and evaluation action, and professionals, the main guarantee of the quality of health actions. The opinions and assessments of patients and professionals have been measured systematically through patient surveys and the quality program for professionals since focusing on the deployment of the successive quality plans [26].

2.3.2. Material and Methods

2.3.2.1. Before describing the consultations and surgical interventions analysed, it would be necessary to provide some information about MC Mutual: what area it covers, what services it provides, whether it is a private or public company, the centres it has, etc., i.e. much of the information below (line 117)

Response: We appreciate the suggestion. Accounting for the given suggestion, the text is modified and a new paragraph is included which provides brief descriptive information about MC Mutual in terms of structure and health services and economic benefits covered by the organisation.

Section: Material and Methods // Page: 3

MC Mutual is a private non-profit organisation, financed by the social security system that covers the needs of more than 177,000 companies and 1,500,000 workers. The study considered professional consultations (health care) and 2,983 surgical interventions during the 2017-2019 period. At this point, it has there were 857 health professionals, two hospitals (Clínica Copérnico and Clínica Londres) with 75 beds in total, and 100 outpatient centers, 88 of which deal with accidents at work and occupational diseases, as well as assessing common diseases that receive social security benefits, it conducts an were considered in the evaluation.  Likewise, the study included the management of two insurance hospital centers (Valencia and Bilbao). It has agreements with other entities for its hospital centers or subsidized care centers. In addition, healthcare financing is through centers of other insurance, private and public health service centers, and transportation. This insurance company has some 900 health providers throughout Spain to offer this complementary care. In 2013, MC Mutual launched a Health Quality Plan, which included patient safety as one of its priorities. The plan has been implemented up to the present, considering two main protagonists: patients, as the center of care and evaluation action, and professionals, the main guarantee of the quality of health actions. The opinions and assessments of patients and professionals have been measured systematically through patient surveys and the quality program for professionals since focusing on the deployment of the successive quality plans [26].

2.3.2.2. Are the hospitals in Valencia and Bilbao? If so, relate to the following sentence (line 118)

Response: We appreciate the need of explanation. Accounting for the given request we must say that MC Mutual hospitals are Clínica Copérnico and Clínica Londres, which are named in the new paragraph (please refer to suggestion 2.3.2.1.). The hospitals in Valencia and Bilbao are shared with other mutual collaborators with social security. To avoid confusion, reference is only made to MC Mutual's own hospitals, so the hospitals of Valencia and Bilbao have been removed from the text.

Section: Material and Methods // Page: 3

MC Mutual is a private non-profit organisation, financed by the social security system that covers the needs of more than 177,000 companies and 1,500,000 workers.The study considered professional consultations (health care) and 2,983 surgical interventions during the 2017-2019 period. At this point, it has there were 857 health professionals, two hospitals (Clínica Copérnico and Clínica Londres) with 75 beds in total, and 100 outpatient centers, 88 of which deal with accidents at work and occupational diseases, as well as assessing common diseases that receive social security benefits, it conducts an were considered in the evaluation.  Likewise, the study included the management of two insurance hospital centers (Valencia and Bilbao). It has agreements with other entities for its hospital centers or subsidized care centers. In addition, healthcare financing is through centers of other insurance, private and public health service centers, and transportation. This insurance company has some 900 health providers throughout Spain to offer this complementary care. In 2013, MC Mutual launched a Health Quality Plan, which included patient safety as one of its priorities. The plan has been implemented up to the present, considering two main protagonists: patients, as the center of care and evaluation action, and professionals, the main guarantee of the quality of health actions. The opinions and assessments of patients and professionals have been measured systematically through patient surveys and the quality program for professionals since focusing on the deployment of the successive quality plans [26].

2.3.2.3. (line 144)

Response: We appreciate the suggestion, but we have not been able to detect the content of this comment.

2.3.2.4. It is not clear whether the questionnaires were administered in person or by telephone (line 146)

Section: Material and Methods // Page: 4

Since 2016, patients' opinions have been assessed annually through satisfaction surveys incorporating the patient's voice in evaluating care experience. The first surveys measured the satisfaction of patients treated for professional consultations. For inpatients, the evaluation was administered physically to the patient at the time of medical discharge. The aspects evaluated were the information received, treatment, waiting time, and facilities. For outpatients, the surveys were administered to patients (for outpatient care) by telephone shortly after being seen. The care modalities considered were: Professional Contingency (CP), Common Contingency (CC), Rehabilitation (RHB), and Emergency (URG). In these surveys, the evaluation of traditional aspects was collected, including information received and treatment, and information incorporated on flexibility and adaptation, conformity with diagnoses and treatments, the comfort of the facilities, accessibility and proximity of the centers and agility, and times of wait in services. At the clinics, patients with or without indication for hospital admission underwent surveys during the discharge process, which were administered physically. In the case of the outpatient centres, the assessment of work-related care is used.At the clinics, the services that the patients evaluated are Hospitalization, Surgery, Rehabilitation, External Consultations, and Psychology, with the same dimensions as in outpatient centers. The survey response scale scores one to ten, wherein one means very dissatisfied, and ten means very satisfied [26]. For this study, The years 2017, 2018, and 2019 have been selected, within which the surveys were carried out in healthcare centers.

2.3.2.5. How many items does this scale have? How do you calculate the total score? Has this survey got different dimensions? (line 155)

Response: We appreciate the need of explanation. Accounting for the given request we must say that the calculation of the scores, the scale used and the dimensions considered from the surveys are answered by referencing the paragraph with reference number 26.

Section: Material and Methods // Page: 3

(…) In 2013, MC Mutual launched a Health Quality Plan, which included patient safety as one of its priorities. The plan has been implemented up to the present, considering two main protagonists: patients, as the center of care and evaluation action, and professionals, the main guarantee of the quality of health actions. The opinions and assessments of patients and professionals have been measured systematically through patient surveys and the quality program for professionals since focusing on the deployment of the successive quality plans [26].

2.3.2.6. What does 'adapted questions' mean? (line 165)

Section: Material and Methods // Page: 5

Surveys were conducted using adapted questions on five successive health quality plan consultations (to assess the level of development of the quality initiatives defined in each of the plans).

2.3.2.7. How many quality referents work in the mutual? (line 172)

Response: We appreciate the suggestion. Accounting for the given suggestion, we have incorporated the number of Referents.

Section: Material and Methods // Page: 5

2.3.2.8. Which results? (line 198)

Response: We appreciate the need of explanation. Accounting for the given request we must say that the concordance of rankings between internal and external experts is referred to above.

Section: Material and Methods // Page: 6

With the scores results obtained by external experts, the Kendall Coefficient of Concordance (W) [22 27] was calculated.

2.3.3. Results

2.3.3.1. In the section on Material and Method 857 professionals are listed. It is not clear where these percentages come from. (line 213)

Response: We appreciate the suggestion. Accounting for the given suggestion, the number of health professionals surveyed per year is clarified.

Section: Material and Methods // Page: 5

The Quality Antenna survey collected the opinion of MC MUTUAL professionals from both large centers (with more than ten health professionals and more than 5,000 medical attentions per year) and those considered small centers (with less than four professionals in health services and less than 1,500 medical attentions per year), which during the period ranged between 150 and 170 health professionals.

2.3.3.2. You should explain in methods what are the criteria for moderate, good, etc... degree of agreement (line 220)

Response: We appreciate the need of explanation. Accounting for the given request we must say that according to Landis & Koch agreement classification, fair agreement values of 0.20 to 0.40 and moderate agreement values of 0.40 to 0.60 are considered, and has been inserted in the text.

Section: Material and Methods // Page: 7

(…) The Kendall Coefficient result was interpreted using the classification established by Landis and Koch [28] to describe the consistency associated with the statistic with a range of agreements ranging from poor to perfect.

2.3.3.3. It is not clear from which questionnaires these items have been removed (line 226)

Response: We appreciate the need of explanation. Accounting for the given request we must say that the items that have been removed belong to the Quality Antenna questionnaire for professionals and referents and this clarification has been inserted in the text.

Section Results // Page 7

(…) Specific categories, such as treatment referrals, lack of continuity of care, and information overload, have been eliminated from the surveys (to referents and professionals) since there was low agreement between the experts the results were consolidated.

2.3.3.4. The analysis of referents is not one of the objectives of the study. Should it not be included as an objective? (line 228)

Response: We appreciate the need of explanation. Accounting for the given request we must say that the referents are not the target of our study and are discarded as an element of comparison as they show perfectly correlated values with the rest of the health professionals.

2.3.3.5. This analysis method has not been explained in methods section (line 243)

Response: We appreciate the suggestion. Accounting for the given suggestion, we have incorporated the analysis method description.

Section: Material and Methods // Page: 6

Finally, to establish the comparison, the following steps were taken. First, to standardize the scores of the two surveys on a scale of 1 to 5 points, a regression line was calculated that allowed the scores to be transformed and made comparable (Y: 5/9+4/9*X). Data from both surveys were on the same scale and integrated the scores into a summary table of the three groups. We used diachronic analyses to formulate the evolutionary trend of each group of patients, professionals, and referents. Later, an order was assigned to the average of the scores of professionals, referents, and patients, allowing the comparative analysis. Spearman's ordinal regression coefficient has been used to compare the ordinal values of professionals and referents first, then to compare professionals (including referents) and patients. Finally, the ordinal values were presented (the first is the best and higher numbers are worse) and a graph of quadrant coincidences and divergences between the professionals (including the referents) and the patients was elaborated.

2.3.3.6. This means that there is a significant difference between those groups... these data should be shown in the table. None of this has been explained in methods section (line 243)

Response: We appreciate the suggestion. Accounting for the given suggestion, the degree of significance of spearman's correlation is clarified by introducing a new sentence.

Section: Results // Page: 7

The values observed in patients, referents, and professionals obtained in the three years considered were averaged (Table 3). When ordering the values, there are no differences between referents and professionals, which show a very high correlation between their values (Spearman's Rho 0.84, p<0.01).

2.3.3.7. What are the criteria for good or bad? Not explained in the Methodology section. (line 260)

Response: We appreciate the suggestion. Accounting for the given suggestion, the value of ordinals is clarified in Material and Methods.

Section: Material and Methods // Page: 6

Finally, to establish the comparison, the following steps were taken. First, to standardize the scores of the two surveys on a scale of 1 to 5 points, a regression line was calculated that allowed the scores to be transformed and made comparable (Y: 5/9+4/9*X). Data from both surveys were on the same scale and integrated the scores into a summary table of the three groups. We used diachronic analyses to formulate the evolutionary trend of each group of patients, professionals, and referents. Later, an order was assigned to the average of the scores of professionals, referents, and patients, allowing the comparative analysis Spearman's ordinal regression coefficient has been used to compare the ordinal values of professionals and referents first, then to compare professionals (including referents) and patients. Finally, the ordinal values are presented (the first is the best and higher numbers are worse) and a graph of quadrant coincidences and divergences between the professionals (including the referents) and the patients was elaborated.

2.3.4. Discussion

2.3.4.1. The discussion would be more comprehensible if it were ordered in relation to each of the dimensions assessed in the survey. It would be necessary to compare these data with those available in the literature and reflect on the reasons for these differences/similarities. (line 273)

Response: We appreciate the need of explanation. Accounting for the given request we must say that the commentary on each of the dimensions would possibly order the discussion in a more consistent way, as the referee indicates. We have preferred to highlight the analysis by quadrants of coincidences and divergences as we consider it more relevant to our conclusions and more reasonable based on the methodology used and the local reality considered.  The comments made on other studies reflect the information provided by the bibliography. We have not found similar studies in terms of the analysis by coincidence and divergence of the different dimensions of health quality.

2.3.4.2. PREMS, PROMS... nothing has been said previously .... Points 24, 25 and 26 should be in the introduction rather than in the discussion (line 283)

Response: We appreciate the suggestion. Accounting for the given suggestion, the paragraph referring to PREMS and PROMS is relocated to the Introduction section and corrected the order of the references.

Surveys of patients and professionals can be an element to deepen this analysis and concretize the value we use in our organization. Data on the PREMS (service provided) and PROMS (individual experience) measures provide critical information for developing patient-centered care. They reduce the observed gap between the vision of the physician and the patient, and design treatment plans to meet the patient's preferences and needs [18]. Some surveys allow approximating the PROMS and PREMS with questions addressed to patients and professionals. [19]

2.3.4.3. Less than the patient themselves? This phrase is not understood (line 313)

Response: We appreciate the suggestion. Accounting for the given suggestion, the sentence is modified to make it more understandable.

Section: Discussion // Page: 10

In Korean care centers, different perceptions were detected between nurses and patients. Nurses rated health care worse than patients. (…)

2.3.4.4. Why do you think these differences between studies exist? (line 330)

Response: We appreciate the need of explanation. Accounting for the given request we must say that the hypothesis we consider is that local circumstances, patients and professionals, as well as existing structures, probably play a specific role that requires a specific analysis and is not very generalisable.

2.3.4.5. The data have been analysed in a quantitative, not qualitative way (line 372)

Response: We appreciate the need of explanation. Accounting for the given request we must say that The data comes from systematic surveys obtaining values that are treated as qualitative (good-bad, better-worse, ...) and in other cases summarised in scores.

2.3.5. Conclusions

2.3.5.1. This conclusion does not follow from either the objectives or the results of the study (lines 379 to  382)

Response: We appreciate the need of explanation. Accounting for the given request we must say that the stated objectives may represent an inventive to transparency that is well appreciated by respondents.

2.3.6. Appendix B

2.3.6.1. As they are the same questions, they could be grouped together to simplify the table (line 411)

Response: We appreciate the suggestion. Accounting for the given suggestion, we have merged the two columns (Referents and Professionals) into one. Please refer to Appendix B.

2.3.6.2. As they are the same questions, they could be grouped together to simplify the table (line 411

Response: We appreciate the suggestion. Accounting for the given suggestion, we have merged the two columns (Referents and Professionals) into one. Please refer to Appendix B.

Reviewer 3 Report

This manuscript is essential to the literature regarding the difference in perception of healthcare management between patients and professionals. I enjoyed reading and critically evaluating this manuscript. However, a few points are needed to be considered in this article,

Abstract: This part has properly explained the background and needs of the study. However, the result and conclusion part needs to be revised and should be more specific. Keywords: Simple and understandable, but a few more keywords could be added.

Introduction: The introduction provides a good, generalized background of the topic that quickly gives the reader an appreciation of the requirement of difference in perception of health care management between patients and professionals. However, to make the introduction more substantiated, the authors may add more references related to the need of the study. A few abbreviations need to be addressed, like IHI. In line 57, some studies have been mentioned. However, the authors have just cited one study; please check and correct it. Overall, there is a need to improve the introduction part to understand the need for the study.

Material and method: This part is well-explained and has covered all the parts in three different phases. However, the statistical analysis of data has yet to be explained properly. The authors have only mentioned using diachronic analysis and comparative analysis. But have yet to explain on what bases the three graphs have been formed.

Results: Tables and figures are proper and self-explanatory. The written part, however, should be more elaborated for readers to gain understanding.

Discussion: In this part, authors did not discuss the results obtained in the study and have not supported the results properly. 

The Authors have mentioned the limitation; however, strengths and further recommendations have yet to be mentioned. It needs to be made clear how this study was helpful for policymakers and stockholders. Also how the lacking areas can be focused upon.

Conclusion: This part of the study needs to provide a concrete conclusion as to how this study on the survey will further help policymakers and stakeholders of Madrid hospitals to improve the services provided by the hospital.

References are appropriately marked, and no duplication is seen. 

Author Response

Reviewer 3

This manuscript is essential to the literature regarding the difference in perception of healthcare management between patients and professionals. I enjoyed reading and critically evaluating this manuscript. However, a few points are needed to be considered in this article,

Abstract: This part has properly explained the background and needs of the study. However, the result and conclusion part needs to be revised and should be more specific. Keywords: Simple and understandable, but a few more keywords could be added.

Introduction: The introduction provides a good, generalized background of the topic that quickly gives the reader an appreciation of the requirement of difference in perception of health care management between patients and professionals. However, to make the introduction more substantiated, the authors may add more references related to the need of the study. A few abbreviations need to be addressed, like IHI. In line 57, some studies have been mentioned. However, the authors have just cited one study; please check and correct it. Overall, there is a need to improve the introduction part to understand the need for the study.

Material and method: This part is well-explained and has covered all the parts in three different phases. However, the statistical analysis of data has yet to be explained properly. The authors have only mentioned using diachronic analysis and comparative analysis. But have yet to explain on what bases the three graphs have been formed.

Results: Tables and figures are proper and self-explanatory. The written part, however, should be more elaborated for readers to gain understanding.

Discussion: In this part, authors did not discuss the results obtained in the study and have not supported the results properly. 

The Authors have mentioned the limitation; however, strengths and further recommendations have yet to be mentioned. It needs to be made clear how this study was helpful for policymakers and stockholders. Also how the lacking areas can be focused upon.

Conclusion: This part of the study needs to provide a concrete conclusion as to how this study on the survey will further help policymakers and stakeholders of Madrid hospitals to improve the services provided by the hospital.

References are appropriately marked, and no duplication is seen. 

3.1. This manuscript is essential to the literature regarding the difference in perception of healthcare management between patients and professionals. I enjoyed reading and critically evaluating this manuscript.

Response: We are very grateful for your positive feedback.

3.2. However, a few points are needed to be considered in this article,

3.2.1. Abstract:

3.2.1.1. This part has properly explained the background and needs of the study. However, the result and conclusion part needs to be revised and should be more specific.

Response:

We appreciate the suggestion. The journal guidelines recommend 200 words in the abstract. We have included information in the section on results and conclusions that we believe can better close the contributions of this study.

Abstract: Patient perception and the organizational and safety culture of health professionals are an indirect indicator of the quality of care. Both patient and health professional perceptions were evaluated, and their degree of coincidence was measured in the context of a mutual insurance company (MC Mutual). This study was based on the secondary analysis of routine data available in databases on patients' perceptions and professionals' evaluations of the quality of care provided by MC Mutual during the period of 2017-2019, prior to the COVID-19 pandemic. Eight dimensions were considered: results of care, coordination on professionals, trust-based care, clinical and administrative information, facilities and technical means, confidence in diagnosis, treatment and confidence in treatment. Patients and professionals agreed on the dimension of good confidence in treatment, as well as on and poor coordination and confidence in diagnosis. They diverged in on confidence and in treatment, which were rated worse by patients than by professionals, and in on results, information and infrastructure, which were rated worse by professionals only. The care managers have to reinforce the training and supervision activities of the positive coincident aspects (therapy) for their maintenance, as well as the negative coincident ones (coordination and diagnostic) for the improvement of both perceptions. The review of patient and professional surveys is a very useful element for the supervision of the health quality of the actions in the context of an occupational mutual insurance company.

3.2.1.2. Keywords: Simple and understandable, but a few more keywords could be added.

Response:

Response: We appreciate the suggestion. We have added the following MesH keywords and terms:

Section: Keywords// Page: 1

Patient Outcome Assessment, Patient-Centered Care, Mutual Insurance Company, Health Care Economics and Organizations

3.2.2. Introduction:

3.2.2.1. The introduction provides a good, generalized background of the topic that quickly gives the reader an appreciation of the requirement of difference in perception of health care management between patients and professionals. However, to make the introduction more substantiated, the authors may add more references related to the need of the study.

Response:  Please refer to suggestions 2.3.1.3. y 2.3.1.4. from Reviewer 2.

3.2.2.2. A few abbreviations need to be addressed, like IHI.

Response:  Please refer to suggestion 2.3.1.1. from Reviewer 2.

3.2.2.3. In line 57, some studies have been mentioned. However, the authors have just cited one study; please check and correct it.

Response: Please refer to suggestion 2.3.1.2. from Reviewer 2.

3.2.2.4. Overall, there is a need to improve the introduction part to understand the need for the study.

Response: We appreciate the suggestion. Accounting for the given suggestion, the introduction has been improved according to Reviewer 2 suggestions.

3.2.3. Material and method:

3.2.3.1. This part is well-explained and has covered all the parts in three different phases.

Response: We are very grateful for your positive feedback.

3.2.3.2. However, the statistical analysis of data has yet to be explained properly. The authors have only mentioned using diachronic analysis and comparative analysis. But have yet to explain on what bases the three graphs have been formed.

Response: Please refer to suggestions 2.3.2.5., 2.3.2.6., 2.3.3.2., 2.3.3.4., 2.3.3.5., 2.3.3.6. and 2.3.3.7. from Reviewer 2.

3.2.4. Results:

3.2.4.1. Tables and figures are proper and self-explanatory.

Response: We are very grateful for your positive feedback.

3.2.4.2. The written part, however, should be more elaborated for readers to gain understanding.

Response: We appreciate the suggestion. Accounting for the given suggestion, the whole text has been improved according to all Reviewers´ suggestions.

3.2.5. Discussion:

3.2.5.1. In this part, authors did not discuss the results obtained in the study and have not supported the results properly. 

Response: We appreciate the need of explanation. Accounting for the given request we must say that the comments made on other studies reflect the information provided in the literature. We have not found similar works in terms of coincidence and diverging analysis of the different dimensions of health quality.

3.2.5.2. The Authors have mentioned the limitation; however, strengths and further recommendations have yet to be mentioned. It needs to be made clear how this study was helpful for policymakers and stockholders. Also how the lacking areas can be focused upon.

Response: We appreciate the need of explanation. We have mainly included the strengths that we have found after the category analysis

Section: Discussion (limitations)// Page: 12

The data come from a labor mutual and must be considered typical of this environment, so comparisons must be used with caution. However, myriad studies in different environments are mentioned in the work.

Most mutuals collaborating with Social Security systematically carry out opinion studies of patients and professionals, which invites us to confirm the conclusions in other entities like ours before concluding on their applicability and generalization.

The data comes from systematic surveys; therefore, it is eminently qualitative data and comparisons. A transformation of the data has been used to make the categories of analysis comparable. An attempt has been made to minimise subjectivity by having a panel of experts show their degree of agreement on the proposed categorisation.

The data refer to the pre-pandemic period, so the situation should be reviewed once the episode pandemic is over.

Despite the limitations of the study, it is worth highlighting the strengths which have been found: firstly, there is a minimum set of categories which can compare two points of view of two types of stakeholders in a healthcare organisation. On the other hand, this is a pioneering study as far as this type of organisation is concerned, providing aspects which give visibility to the elements which should be incorporated into future strategic plans in terms of Quality and Patient Safety, from two main actors in the care process.

3.2.6. Conclusion:

3.2.6.1. This part of the study needs to provide a concrete conclusion as to how this study on the survey will further help policymakers and stakeholders of Madrid hospitals to improve the services provided by the hospital.

Response: We appreciate the suggestion. Accounting for the given suggestion, we have incorprated a new paragraph in the text in order to improve the conclusions.

Section: Conlusions // Page: 12

The care managers have to reinforce the training and supervision activities of the positive coincident aspects (therapy) for their maintenance, as well as the negative coincident ones (coordination and diagnostic) for the improvement of both perceptions.

3.2.7. References are appropriately marked, and no duplication is seen. 

Response: We are very grateful for your positive feedback.

Reviewer 4 Report

During the period of 2017-2019, how many articles and data were reviewed, and based on the entry conditions of all data, how many studies were excluded?

The introduction and necessity of conducting the study needs to be expanded.

Author Response

Reviewer 4

During the period of 2017-2019, how many articles and data were reviewed, and based on the entry conditions of all data, how many studies were excluded?

The introduction and necessity of conducting the study needs to be expanded.

4.1. During the period of 2017-2019, how many articles and data were reviewed, and based on the entry conditions of all data, how many studies were excluded?

Response: We appreciate the need of explanation. Accounting for the given request we must say that as we have not intended to make an exhaustive bibliographical review, we have collected the scientific papers that we have considered appropriate.

4.2. The introduction and necessity of conducting the study needs to be expanded.

Response: We appreciate the suggestion. Accounting for the given suggestion, we have improved the introduction section according to all Reviewers´ suggestions.

Round 2

Reviewer 3 Report

The authors have covered all the desired points and made the manuscript easy to understand for readers.

Reviewer 4 Report

The relevant bugs were identified